# A Human Health Risk Assessment of Trace Elements Present in Chinese Wine

**DOI:** 10.3390/molecules24020248

**Published:** 2019-01-11

**Authors:** Zhi-Hao Deng, Ang Zhang, Zhi-Wei Yang, Ya-Li Zhong, Jian Mu, Fei Wang, Ya-Xin Liu, Jin-Jie Zhang, Yu-Lin Fang

**Affiliations:** 1College of Enology, Northwest A&F University, Yangling 712100, China; 13389254907@163.com; 2Inspection and Quarantine Technology Center of Qinhuangdao Entry-Exit Inspection and Quarantine Bureau, Qinhuangdao 066004, China; muyishixinrenwei@163.com (Z.-W.Y.); zhongyali@163.com (Y.-L.Z.); mujian@163.com (J.M.); wangfei@163.com (F.W.); liuyaxin@163.com (Y.-X.L.); zhangjinjie@163.com (J.-J.Z.)

**Keywords:** trace elements, health risk assessment, estimated daily intake, ICP-MS, Chinese wine

## Abstract

The concentrations of trace elements in wines and health risk assessment via wine consumption were investigated in 315 wines. Samples were collected from eight major wine-producing regions in China. The concentrations of twelve trace elements were determined by inductively coupled plasma mass spectrometry (ICP-MS) and Duncan’s multiple range test was applied to analyze significant variations (*p* < 0.05) of trace elements in different regions. Based on a 60 kg adult drinker consuming 200 mL of wine per day, the estimated daily intake (EDI) of each element from wines was far below the provisional tolerable daily intake (PTDI). Health risk assessment indicated the ingestion influence of individual elements and combined elements through this Chinese wine daily intake did not constitute a health hazard to people. However, Cr and Mn were the potential contaminants of higher health risk in Chinese wines. The cumulative impact of wine consumption on trace elements intake in the daily diet of drinkers should not be ignored due to the presence of other intake pathways.

## 1. Introduction

In recent years, wine has become increasingly popular in China, and the production of Chinese wine gradually cannot be underestimated worldwide. Compared with the world wine industry, the Chinese wine industry started late but developed rapidly. In 2016, the production of Chinese wine was 1.14 million liters, ranking sixth in the world [1]. On the other hand, the area of Chinese vineyards is extremely large, totaling 864 thousand hectares in 2016 to rank second in the world [2], so Chinese wines are widely distributed in China. The main areas producing Chinese wine are the eastern foothills of Helan Mountain (HM), the Hexi Corridor (HC), Xinjiang (XJ), the ring around the Bohai Gulf (BG), the Loess Plateau (LP), Southwest Highland (SWH), the Northeast (NE) and Yanhuai Valley (YV). The Helan Mountain region is the most important for the Chinese wine industry and is also the major source of high-quality wine in China. Due to its latitude similarity with Bordeaux, it is also called the Bordeaux of China. The Hexi Corridor region is located in Gansu Province included in the Silk Road Economic Belt proposed by China. The environment here is similar to a cool semidesert that is beneficial to the development of grape aromas responsible for wine flavors attributes. The Xinjiang region is not only the largest Chinese wine-producing area, but the grapes also have high sugar accumulation because winemakers delay harvesting to ensure a fully development of aromas and accompanied by dry climate, so the alcohol content of wine is generally high. The Bohai Gulf region is the earliest wine-producing area in China, and it is the birthplace of many old wine enterprises and has a temperate monsoon climate. The Loess Plateau region is located in the middle part of China, with a continental monsoon climate, and has a long history of viticulture. The Southwest Highland region lies in the Hengduan Mountain area of southwestern China, the climate is similar to an alpine valley with the complex landforms. Because of this region’s high accumulated temperature and sufficient light, it is also suitable for grape growth. The Northeast region is situated in Northeast China, where latitude is higher and climate is colder. The temperature here is commonly unfit for most grapes except the amur grape, and thus, the wine produced here has unique characteristics such as deep color and the high concentration of minerals and tannins. The Yanhuai Valley region is a burgeoning wine-producing region in China mainly located in the Huaizhuo basin of northwest Beijing, the unique valley ecological climate here is a paradise for grape growth.

Although wine has been increasingly proven to be beneficial to human health due to the inclusion of a variety of nutrients such as amino acids, vitamins, mineral elements, and polyphenols [3,4,5,6,7], no research has encouraged people to drink wine. In addition to restrictions on alcohol, there is a lack of comprehensive assessment of wine safety risks. Food safety has always been a focus of public attention worldwide, particularly in China. Recently, related issues have increasingly been focalized on government and public opinion. With the destruction and pollution of the environment, some of the pollutants have indirectly emerged in our food. The Joint FAO/WHO Expert Committee on Food Additives (JECFA) reports the tolerance limits for human intake of some pollutants. Arsenic is a trace element and also a high-risk carcinogen with a low lethal dose to humans [8], and the provisional tolerable weekly intake (PTWI) of arsenic is 15.0 μg/kg body weight [9]. Lead is a common toxic metal element that affects the hematological, nervous and reproductive systems of humans and causes pathological changes in organs, leading to a decline in the intelligence quotient (IQ) for children [10]. Cadmium is a nonessential element for the human body but is toxic to the kidneys, bones and cardiovascular system [11], and the provisional tolerable monthly intake (PTMI) of cadmium is 25.0 μg/kg body weight [12]. Chromium is an essential element of the human body, nevertheless, it also harms the human body depending on its valence state. Trivalent chromium is a beneficial essential element, but the toxicity of hexavalent chromium is serious for the human body [13]. Nickel is not an essential trace element, it can cause a variety adverse effects of pulmonary, carbonyl nickel has acute toxicity and carcinogenicity [14], and the provisional tolerable daily intake (PTDI) of nickel based on lowest observed adverse effect level (LOAEL) is 12.0 μg/kg body weight [15]. The above contaminants, especially trace elements, have been detected in previous reports from wine [16]. Although the poisoning caused by drinking wine has not been reported directly, these contaminants are still destructive to the human body due to their chronic and persistent effects. In addition, some beneficial elements in wine such as manganese, cobalt, copper, zinc, molybdenum, aluminum and selenium can also cause adverse reactions if their intake exceeds a specific health indicator. For instance, molybdenum is an indispensable trace element for organisms, but the high concentration of molybdenum intake negatively affects semen quality [17]. The provisional maximum tolerable daily intake of zinc is 1.00 mg/kg body weight [18], and the PTWI of aluminum is 1000 μg/kg body weight [19]. The upper tolerable limits of daily intake for manganese and selenium are 11.0 mg/day [20] and 400 μg/day [21] by a 60 kg adult, respectively. Because compared with other routes, including inhalation and dermal contact, food and drink consumption have been suggested to be a major source of human exposure of trace elements [22,23], we must consider the potential impact of these trace elements from wine on the daily diet of drinkers.

Nevertheless, publications on the concentrations of some adverse trace elements in Chinese wines and health risk assessments of the dietary ingestion of these elements from wine are scant. On the other hand, several agencies and organizations such as the US Environmental Protection Agency (US EPA) and JECFA have provided guidelines on the intake of trace elements by humans. Hence, it is necessary to estimate the health risk of Chinese wine from different wine-producing regions. The main objectives of the present study were: (1) to establish and validate an analytical method for the determination of trace elements in wines by inductively coupled plasma mass spectrometry (ICP-MS) after microwave-assisted digestion; (2) to determine the concentrations of aluminum, arsenic, cadmium, chromium, cobalt, copper, lead, manganese, molybdenum, nickel, selenium and zinc in wines produced from eight different regions of China; (3) to compare the determination levels of different regions with the results reported in previous literature and to analyze the distribution characteristics of trace elements in Chinese wines; and (4) to estimate the dietary intake rates of some elements in Chinese wines through the daily consumption of wine and the health risk of trace elements intake at these rates for Chinese wine.

## 2. Results and Discussion Sections in Wrong Order–Experimental is Last–Renumber Things Affected

### 2.1. Method Verification Results

The validation results of the ICP-MS method established in this study for trace elements in wine were shown in the Table 1. The correlation coefficient of calibration curves established by mixing standard solutions with different gradients were 0.983–0.999. According to the IUPAC method, continuous parallel determination of 2% HNO_3_ blank solution for 11 times, the limit of detection (LOD) of each element was obtained by the 3-fold standard deviation converting to concentration by calibration curves, and the limit of quantitation (LOQ) of each element was obtained by the 10-fold standard deviation converting to concentration by calibration curves. It could be seen that the range of all elements LOD was 0.024–2.62 μg/L, and the range of all elements LOQ was 0.080–8.06 μg/L. Then, the range of precision for all elements obtained by the relative standard deviations of the above 11 times determined reagent blanks was 0.602–4.81%. For the spiked samples of different concentration levels, the range of spike recovery was 81–137%. The determination results of certificated reference materials (GBW 10010) were all within the error range of the certificate values except for those not detected. It indicated for these that the established method was accurate, effective and suitable for the determination of trace elements in wine.

### 2.2. Concentrations of Trace Elements in Wine

The concentration range and distribution of twelve elements in 315 wines from eight different regions determined using ICP-MS were shown in boxplot (Appendix A). As for all wines, the element with the highest average concentration was Mn (3101 μg/L), the element with the lowest average concentration was Cd (0.568 μg/L), and other elements concentrations decreased in the order of Al > Zn > Cu > Cr > Ni > Pb > Se > As > Co > Mo. Irrespective of the origin of the wines, the concentration ranges (in μg/L) of the twelve elements analyzed in all wines were as follows: Cr (76–337), Mn (863–16026), Co (1.15–15.3), Ni (9.67–189), Cu (13.7–543), Zn (77–1670), As (1.54–34.9), Mo (0.390–9.62), Cd (0.076–2.36), Pb (2.06–44.8), Al (90–3202) and Se (6.06–15.1). There were some outliers in the determination results of each element except Se. The structures of the data for the concentrations of each element in study regions were different.

An investigation of the concentrations of trace elements in wine from different regions found that each element was present in different concentrations and distributions in different regions, as shown in Table 2. The measured concentration distribution of Cr showed that all regions had basically the same concentration except the Northeast region, and its average concentration (211 μg/L) was significantly higher than those of the others. The Ni concentration was significantly higher in the Northeast (average 81 μg/L) and Bohai Gulf (average 73 μg/L) regions than in the other regions. The concentration of As was the highest in the Northeast (average 12.2 μg/L) region, which was significantly higher than the concentration in the Xinjiang (average 6.21 μg/L) and Bohai Gulf (average 6.70 μg/L) regions, followed by the Helan Mountain (average 5.94 μg/L), Hexi Corridor (average 5.86 μg/L), Loess Plateau (average 4.02 μg/L), Yanhuai Valley (average 4.43 μg/L) and Southwest Highland (average 4.33 μg/L) regions, whose As concentrations were not significantly different. The same concentration levels of Cd were found in the Xinjiang (average 0.205 μg/L), Helan Mountain (average 0.334 μg/L), Hexi Corridor (average 0.354 μg/L) and Yanhuai Valley (average 0.285 μg/L) regions, and they were the lowest concentrations of Cd among the tested regions, while the highest concentration level was in the Northeast (average 1.35 μg/L) region. The wine of the Xinjiang region had the lowest Pb concentration (average 6.86 μg/L) of all wines, and the wines of Helan Mountain, Hexi Corridor, Yanhuai Valley and Southwest Highland regions had the same Pb concentration level which was higher than Xinjiang region. The Pb concentrations of wine in Northeast, Loess Plateau and Bohai Gulf regions were three higher levels which order was Loess Plateau (average 17.7 μg/L) < Northeast (average 20.6 μg/L) < Bohai Gulf (average 25.2 μg/L). The wine Mn concentrations of the Xinjiang, Helan Mountain, Hexi Corridor, Loess Plateau, Yanhuai Valley and Southwest Highland regions ranged from 863–4780 μg/L, and they were significantly lower there than in the Northeast (average 7807 μg/L) and Bohai Gulf (average 6234 μg/L) regions. The Co concentrations of wine barely did not differ between regions, according to an ANOVA, the Co concentration levels of these regions was ordered Northeast > Loess Plateau and Bohai Gulf > Hexi Corridor > Xinjiang, Helan Mountain and Yanhuai Valley > Southwest Highland. The highest Cu concentration of wine was in the Xinjiang region (average 220 μg/L) but the lowest level was in Yanhuai Valley (average 94 μg/L) and Bohai Gulf (average 97 μg/L) regions. The Zn concentration was divided into four levels: the Northeast region had the highest level (average 724 μg/L), the Bohai Gulf (average 597 μg/L) and Loess Plateau (average 534 μg/L) regions had the second highest level, which was followed by the Helan Mountain (average 378 μg/L), Hexi Corridor (average 394 μg/L), Yanhuai Valley (average 424 μg/L) and Southwest Highland (average 488 μg/L) regions, the last one was in Xinjiang (average 288 μg/L) region. The concentration of Mo was lower in wine in general, and the wine Mo concentration in the Xinjiang region (average 3.43 μg/L) was significantly higher than that in the other regions, moreover, these other regions were approximately in same. The lowest Al concentration was in the Southwest Highland (average 553 μg/L) region, while the highest levels were in the Northeast (average 1058 μg/L) and Loess Plateau (average 1055 μg/L) regions, in addition, the other regions were in same level. In general, the overall Se concentrations of the Northeast (average 12.7 μg/L) region was higher, and that of the other regions were similar to each other.

For the determination of these elements in wine, there have been some previous studies that could be used for comparison. Fiket et al. [24] reported that the concentration ranges of Cr, Ni, As and Cd in wines from eastern Croatia were 6.50–31.1 μg/L, 15.3–50.0 μg/L, 0.690–19.1 μg/L and 0.175–1.88 μg/L, respectively. Sperkova and Suchanek [25] showed that the Cr, Ni, As and Pb in Bohemian wines were 18.0–32.0 μg/L, 15.0–53 μg/L, 1.50–6.70 μg/L and 11.0–48.0 μg/L, respectively. The concentration ranges of Cr and As were 20.0–50.0 μg/L and 0.040–0.800 μg/L, respectively, which were studied in Nebbiolo-based wines by Marengo and Aceto [26]. Gremaud et al. [27] quantified elements such as Mn, Zn and Al in their research, and the concentrations of Mn, Zn and Al ranged from 270–600 μg/L, from 340–1140 μg/L and from 180–1100 μg/L, respectively. Geana et al. [28] analyzed the geographical origins of Mn, Co, Cu and Zn in Romanian wines, and the average concentrations of Mn, Co, Cu and Zn were 806 μg/L, 4.35 μg/L, 501 μg/L and 434 μg/L, respectively. Alkış et al. [29] investigated the concentrations of Mn (average 399 μg/L), Co (average 3.37 μg/L), Cu (average 145 μg/L) and Zn (average 1244 μg/L) in Turkish wines. According to comparison, the concentrations of As, Cd and Co in most studies were similar to those of this study, but the concentrations of Mn and Cr in most studies were lower than this study. On the other hand, the highest concentration of As in this study (34.9 μg/L) was far lower than the International Organization of Vine and Wine (OIV) limit of arsenic in wine (200 μg/L) [30]. The highest concentration of Pb in this study was 44.8 μg/L, which was lower than the OIV limit on lead content in wine (150 μg/L) [30], and it was also lower than the People’s Republic of China national standard for lead content in wine (0.200 mg/kg) [31]. In addition, the OIV issued the highest limit on Cu in wine (1000 μg/L) [30], and the highest Cu concentration (543 μg/L) in this study was lower than this limit. Similarly, the highest concentration of Zn (1170 μg/L) in all the wines was far less than the OIV limit of Zn in wine (5000 μg/L) [30].

### 2.3. Analysis of the Characteristics and Source of Trace Elements in Chinese Wine

The eight regions which wines were selected in this study were the main wine-producing regions in China, so the wines in this study essentially represented all Chinese wine. In summary, the different concentrations of each element in the wine of different regions reflected the diversities of environmental condition and element distribution in these regions. The element concentrations of wine of the Northeast region were relatively high. The main reason for these high levels was that the wines selected in the Northeast region contained very sweet wine such as ice wine. The concentrations of all the components in these wines were increased by concentrating the wines. On the other hand, the Northeast area was an old industrial base in China with abundant mineral resources. Because of the development of industry, the soil in these areas such as the cities of Tonghua, Linjiang and Benxi was rich in various trace elements. The grapes grown here and the wine made with these grapes thus contained greater amounts of trace elements or might be contaminated. The concentration distributions of each element except Cu in the wines of the Helan Mountain and Hexi Corridor regions are very close. First, the geographical position of eastern foothills of Helan Mountain and Hexi Corridor were close, so the differences in the concentration levels of the various elements due to geographical origin were very small. Then, the concentration of most elements in these two regions indicated that their wines were less polluted by trace elements than the wines of other regions. However, their difference in Cu content might be due to the use of different grape cultivation methods such as the use of Bordeaux. Because the main component of Bordeaux solution was basic cupric sulphate, and the demand of Bordeaux solution for vineyards varied from region to region, so the different amount of Bordeaux solution used in different regions led to the difference of copper concentration level. The high concentration levels of elements in the wine from the ring around the Bohai Gulf might indicate that this region was polluted with high concentrations of trace elements, especially Mn, Ni, Cd and Pb. The concentration levels of the most elements in wine were the lowest in the Yanhuai Valley comparing for all the regions. The Cu and Mo concentration levels of wine were significantly higher in Xinjiang region than in the wine of other regions, which was also related to the distribution of local elements, and perhaps they could be the characteristic elements of this region. Finally, the elements in the wines of the Loess Plateau and Southwest Highland did not have a particularly prominent parts in this study. A comparison of the characteristics of the different elements in each region could provide a general description of trace element conditions in Chinese wine.

### 2.4. Estimated Daily Intake of Trace Elements through the Consumption of Wine and Its Health Risk Assessment

To evaluate potential hazards resulting from long-term daily consumption of wine containing these measured elements, we referenced the concept of an estimated daily intake (EDI) for trace elements from wine. As there were no related data about the wine daily consumption rate of drinkers in China, and the target of our assessment was adult drinkers, we assumed that an adult drinker had a daily drinking volume of 200 mL (approximately a glass of wine), which was also similar to the wine consumption rate of 195 g/day used in other assessment study [32]. On the other hand, Chinese drinking habits were based on cups, so the volume of 200 mL wine consumption could also make people more specific understanding about the amounts assessed. Then, because the smaller the body weight, the larger the EDI calculated, we selected 60 kg as the adult drinker average weight for calculations so that the assessment results could be applied to more people. The measured concentration of each element was divided into two parts to evaluate: the used average concentration (mean) represented the general intake, and the used 95% confidence interval upper limit (P95) of the average concentration represented the possible highest intake. Based on the JECFA report of the tolerance limits and perniciousness of pollutants intake, we assessed the EDI of Cr, Mn, Ni, Zn, As, Mo, Cd, Pb, Al and Se from wine of the above eight regions, and the results were shown in Table 3. It could be seen that the EDI (including the P95) of each element in all regions was far lower than the related PTDI. The average EDI of Cr, Mn, Ni, Zn, As, Mo, Cd, Pb, Al and Se for all wines were 0.467, 10.3, 0.124, 1.51, 0.020, 0.007, 0.002, 0.047, 2.70 and 0.032 μg/kg bw/day, respectively. The EDI also basically represented the daily intake per unit weight of these elements through the drinking of 200 mL Chinese wine. For each region, the EDI level was consistent with the concentration level of element. On the other hand, Cr, Mo and Pb did not have clear criteria for the PTDI because of their characteristics. The hexavalent form of chromium was difficult to analyze separately, as chromium (VI) was reduced to chromium (III) in the stomach and gastrointestinal tract [33]. Thus, there were no adequate toxicity studies available to provide a basis for no observed adverse effect level (NOAEL). Molybdenum was ordinarily considered an essential element with an estimated daily requirement of 0.100–0.300 mg for adults [34]. These values were much higher than the EDI obtained in this study, indicating that the intake of Mo from wine hardly affected the intake of Mo in the daily dietary structure. As for lead, because the dose–response analyses did not provide any indication of a threshold for the key effects of lead, JECFA concluded that a guideline of tolerable intake that would be considered to be health protective was not possible to establish [12].

Health risk assessment was the process that evaluated the potential health effects from doses of a contaminant delivered to humans in some manner. The health risks of trace element intake from the consumption of wine were assessed based on the THQ, which had been recognized as a useful parameter for the evaluation of risk associated with a contaminant. The THQ was the ratio of the estimated dose of elements from wines to a corresponding reference dose, and this approach offered an indication of the risk level due to contaminant exposure. Reference doses were obtained by conversion from USEPA data [35], and the reference doses of Cr, Mn, Ni, Zn, As, Mo, Cd and Se were 3.00, 140, 20.0, 300, 0.300, 5.00, 1.00 and 5.00 μg/kg per day, respectively. The estimated THQ values of the elements studied were shown in Table 4, including the THQ at the maximum intake (P95). The THQ values of each estimated element did not exceed 1 in all regions, suggesting that the exposed population would not experience significant health risks when ingesting these individual elements from daily consumption of 200 mL of wine. In addition, the higher THQ values indicated a higher probability of exposed risk, even if the THQ value was not greater than 1.

When all estimated elements were taken into account, the total THQ values performed to estimate the cumulative health risk effect were calculated by sum of THQ values of these elements, which was customarily called the HI. In the table, the HI values of different regions were also less than 1, and the highest HI values were 0.593 and 0.708 (P95), which were found in the Northeast region. Furthermore, the HI values in regions other than the Bohai Gulf region (0.419 and 0.477 for P95) were mostly similar to each other but less than those of the Bohai Gulf region. It showed that these contaminant elements did not cause health hazards to people through the daily consumption of these quantities of wine individually, but the risks that might present were higher in the wine of the Northeast region than in the wine of the other regions. For all the estimated wines, the THQ values of Cr accounted for the highest average proportions of the total THQ values, 48.5% and 47.0% (P95), and the average contribution proportions of Mn to the total THQ values which were 23.8% and 24.2% (P95) ranked second. On the other hand, the THQ also represented the contribution of wine to contaminants in the acceptable range for the daily diet in this study. For example, the average THQ of Cr in the measured wines was 0.159, which meant that the contribution of wine consumption to the tolerable daily intake of Cr was 15.9%. Risk assessment for a specific contaminant intake required comprehensive consideration of all intake pathways, and wine consumption was just one such path. Therefore, the proportion of wine consumption was more important for health risk assessment of wine in the daily diet of drinkers.

## 3. Materials and Methods

### 3.1. Study Area and Sampling

The study area consisted of eight major Chinese wine-producing regions, including the eastern foothills of Helan Mountain (HM), Xinjiang (XJ), the Hexi Corridor (HC), the ring around Bohai Gulf (BG), Southwest Highland (SWH), Yanhuai Valley (YV), the Northeast (NE) and the Loess Plateau (LP) (Figure 1).

Wine samples with vintages all between 2013 and 2014 were collected from a total of 71 local different chateaus or wineries in 2015, which chateaus or wineries had the Protected Geographical Indication (PGI) for wines. All wine samples were taken directly from wine tanks or oak barrels, and the varieties and detailed quantities of analyzed wines are shown in Table 5.

### 3.2. Chemicals and Reagents

Metal-oxide-semiconductor grade concentrated nitric acid (HNO_3_) was obtained from the Beijing Institute of Chemical Reagents (Beijing, China). Deionized water (18.2 MΩ cm) from a Milli-Q system (Millipore, Milford, MA, USA) was used throughout the experiments. A standard solution of trace elements, including aluminum (Al), arsenic (As), cadmium (Cd), chromium (Cr), cobalt (Co), copper (Cu), lead (Pb), manganese (Mn), molybdenum (Mo), nickel (Ni), selenium (Se) and zinc (Zn), at a concentration of 100 mg/L was purchased from Seigniory Chemical Products Ltd. (SCP Science, Montreal, QC, Canada). A mixed tuning solution containing magnesium (Mg), indium (In), cerium (Ce), barium (Ba) and uranium (U) at a concentration of 1.0 μg/L was obtained from PerkinElmer Corporation (Waltham, MA USA). The solution with yttrium (Y) as internal standard was purchased from Seigniory Chemical Products Ltd. Certified rice reference material (GBW 10010) was purchased from the National Standard Substance Research Center (NSSRC, Beijing, China).

### 3.3. Preparation and Analytical Methods

An Anton Paar microwave oven (Anton Paar Multiwave 3000, Anton Paar GmbH, Graz, Austria) was used for all sample digestions. All vessels in the whole study were soaked in 30% HNO_3_ for one night and then rinsed with deionized water more than three times. Five milliliters of wine sample were placed in a PTFE vessel. The vessel was placed on an electric hot plate (LabTech EHD 36, Beijing LabTech Corporation, Beijing, China) at 100 °C to evaporate the ethanol until the wine sample was concentrated to 2 mL volume. Then, 1 mL of concentrated HNO_3_ was added, and the sample was subjected to microwave digestion. The conditions of the microwave-assisted digestion were as follows: (1) temperature: 190 °C; (2) pressure: 200 psi (1 psi = 6890 Pa); (3) climbing time: 20 min; and (4) retention time: 10 min. This procedure was completed in a closed system, and the samples did not contact the outside environment. The colorless transparent liquid in the vessel indicated that digestion was accomplished. When the process finished, the vessel was retained on an electric hot plate at 100 °C to evaporate redundant acid until the sample was concentrated to 2 mL volume. The vessel was cooled to room temperature and then moved into a volumetric flask after cooling. Finally, the digested solution was constant volume to 25 mL with 2% HNO_3_ and waiting for determination. Reagent blanks and matrix spike duplicates were also treated in accordance with the abovementioned procedure.

The element concentrations were determined using ICP-MS (PerkinElmer ICP-MS Elan DRC-e, PerkinElmer Corporation) with a 40.68 MHz self-excited radio frequency generator, GemClean^TM^ cross nebulizer and Ryton^TM^ double channel atomizing chamber of highly inert polymer material. The main optimized instrumental parameters were as follows: (1) radio frequency power: 1100 W; (2) plasma gas flow rate: 15 L/min; (3) carrier gas flow rate: 0.94 L/min; (4) auxiliary gas flow rate: 1.2 L/min; (5) lens voltage: 5.5 V; and (6) sampling flow rate: 0.8 mL/min; (7) detection mode: standard mode. The correction equation recommended by the instrument software was used.

### 3.4. Quality Control

Quality assurance measures comprised analyzing matrix blank samples and the internal standard solution of each batch, and the certified reference material (GBW 10010) were inserted into the sample sequence every 10 samples to verify sensitivity and repeatability. The calibration curve of multi-element mixed standard solution was prepared from 100 mg/L standard solution of trace elements in 2% HNO_3_, where the linear range of Al was 10.0–250 μg/L and the linear ranges of As, Cd, Cr, Co, Cu, Pb, Mn, Mo, Ni, Se, Zn were 0.250–50.0 μg/L. In addition, spike recovery tests were done on wine samples, and the spiked samples were prepared at three different concentration levels. The analytical precision and accuracy were accepted only when relative standard deviation (RSD) values were below 5% for the elements, according to the results of duplicate measurements of all samples and the certificated reference materials.

### 3.5. Statistical Analysis and Calculations

All analyses were performed in triplicate, the mean values of three duplicates were used as the result of each wine analysis. Duncan’s multiple range tests were used to determine significant difference (*p* < 0.05) with SPSS 19.0 software for Windows (SPSS Inc., Chicago, IL, USA). The boxplot was created by SPSS 19.0.

The estimated daily intake (EDI, μg/kg bw/day) of trace elements from wine consumption depended on the concentration of the elements in the wine, daily consumption rate of wine and body weight of the consumers [36]:
EDI = (C × R)/BW(1)
where C—the concentration of an individual element in the wine, μg/L; R—the daily wine consumption rate for adult wine drinkers, L/day; BW—the average body weight of assessed population, kg.

The health risk due to the consumption of wine was assessed based on the target hazard quotient (THQ), which was calculated from the ratio of EDI and an oral reference dose [37]:
THQ = EDI/RfD(2)
where RfD—the oral reference dose for each element, μg/kg bw/day.

If the THQ is less than 1, it means that the intake of this element from wine has no obvious adverse effects. If the THQ is equal to or higher than 1, there is a health risk to humans [38,39]. The total THQ values of the elements evaluated in wine is defined as hazard index (HI) used to evaluate the comprehensive health risks of wine.

## 4. Conclusions

The concentrations of Al, As, Cd, Cr, Co, Cu, Pb, Mn, Mo, Ni, Se and Zn in 315 wines sourced from the eight major Chinese wine-producing regions were determined, and the established method could accurately and effectively analyze the trace elements in wine. The results from this study suggested that the concentration levels of elements measured in wines decreased in the order of Mn > Al > Zn > Cu > Cr > Ni > Pb > Se > As > Co > Mo > Cd, and the concentrations of Mn and Cr were clearly higher than those in previous studies. The Northeast region had higher concentration level of these elements, and the Yanhuai Valley region had lower concentration level on the whole, which reflected the environmental condition and element distribution in these regions, with differences for different regions.

The estimated daily intake of elements with potential health risks from wine consumption, assuming a daily wine consumption of 200 mL for 60 kg drinkers, was far lower than the related tolerable daily intake for this element, including the possible highest intake. The health risk assessment indicated that the THQ of each evaluated element was all far below 1, which meant that the exposed population would not experience significant health risks from daily consumption of 200 mL of wines individually in ingesting these elements. The HI for wine consumption in each region was also less than 1, which again showed that the contribution of wine consumption did not pose a threat to tolerable daily intake of potential risk elements in the daily diet. However, the THQ values of Cr and Mn accounted for a larger proportion of the HI, which meant that they were major contaminants in the consumption of Chinese wine and needed more attention. On the other hand, the proportion of wine consumption in the daily diet of drinkers was more important for comprehensive health risk assessment of contaminants due to the presence of other intake pathways. Therefore, the cumulative impact of wine consumption on trace elements intake in the daily diet of drinkers should not be ignored.

## Figures and Tables

**Figure 1 molecules-24-00248-f001:**
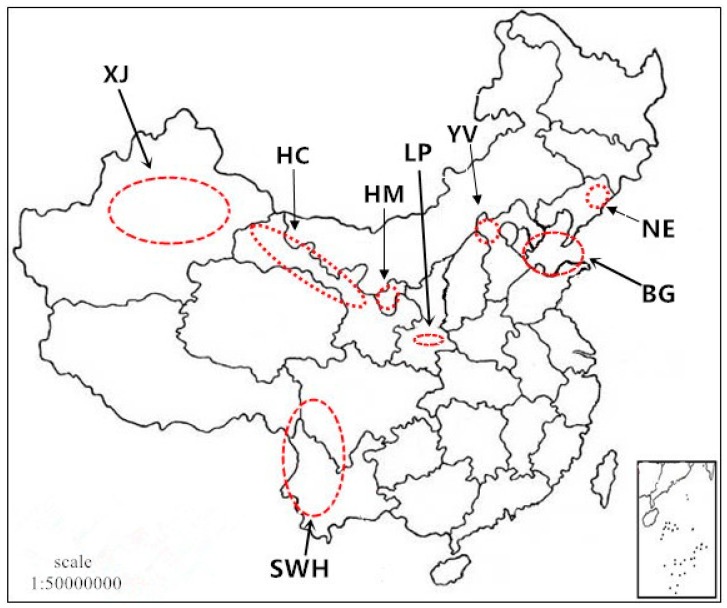
Geographical location of study areas of wine production in China (HM: Helan Mountain; XJ: Xinjiang; HC: Hexi Corridor; BG: Bohai Gulf; SWH: Southwest Highland; YV: Yanhuai Valley; NE: Northeast; LP: Loess Plateau).

**Table 1 molecules-24-00248-t001:** The method verification results for the determination of trace elements in wine by ICP-MS.

Element	Slope	Correlation Coefficient (R^2^)	Linear Range (μg/L)	LOD (μg/L)	LOQ (μg/L)	Precision (%, n = 11)	Added (μg/L)	Recovery (%)	GBW 10010 (mean ± SD, μg/L)
^52^Cr	0.011	0.997	0.250–50.0	0.525	1.73	1.20	50.0	84	ND
							100	85	
							150	91	
^55^Mn	0.016	0.999	0.250–50.0	0.111	0.366	0.602	500	135	17.1 ± 0.724
							1500	137	
							3000	135	
^59^Co	0.013	0.999	0.250–50.0	0.024	0.080	3.31	1.00	100	ND
							5.00	90	
							10.0	89	
^60^Ni	0.003	0.999	0.250–50.0	0.157	0.517	2.60	50.0	83	0.284 * ± 0.046
							100	85	
							150	87	
^63^Cu	0.006	0.999	0.250–50.0	0.060	0.196	2.93	500	91	4.79 ± 0.289
							1500	122	
							3000	115	
^66^Zn	0.002	0.999	0.250–50.0	0.780	2.58	1.52	500	93	24.3 ± 1.65
							1500	93	
							3000	93	
^75^As	0.002	0.999	0.250–50.0	0.167	0.552	0.904	1.00	123	ND
							5.00	118	
							10.0	117	
^98^Mo	0.006	0.999	0.250–50.0	0.025	0.081	1.61	1.00	108	0.515 ± 0.057
							5.00	95	
							10.0	94	
^111^Cd	0.003	0.999	0.250–50.0	0.038	0.124	1.02	0.050	135	0.089 * ± 0.009
							0.100	108	
							0.500	99	
^208^Pb	0.021	0.999	0.250–50.0	0.052	0.171	3.20	1.00	91	0.096 * ± 0.032
							5.00	81	
							10.0	86	
^27^Al	0.008	0.983	10.0–250	1.31	4.33	4.81	500	107	363 ± 21.4
							1500	100	
							3000	90	
^77^Se	0.0002	0.997	0.250–50.0	2.62	8.06	4.53	1.00	111	ND
							5.00	124	
							10.0	122	

Notes: * The value was below to LOQ but higher than LOD as the estimated result. The results larger than the linear range were calculated after dilution. ND = not detected. SD = standard deviation.

**Table 2 molecules-24-00248-t002:** Concentrations (mean ± SD, μg/L) of trace elements in 315 wines of different regions.

Element	NE (n = 30)	XJ (n = 50)	HM (n = 60)	HC (n = 30)	LP (n = 20)	YV (n = 35)	BG (n = 40)	SWH (n = 50)
Cr	211 ± 94d	138 ± 27.1abc	131 ± 28.8abc	140 ± 26.0bc	133 ± 25.5abc	123 ± 22.7ab	151 ± 24.1c	118 ± 33.3a
Mn	7807 ± 3786c	1828 ± 471a	1868 ± 491a	1964 ± 500a	2228 ± 431a	2060 ± 612a	6234 ± 3116b	2283 ± 751a
Co	8.01 ± 2.80d	3.29 ± 1.22ab	3.55 ± 1.22ab	4.04 ± 2.03b	5.05 ± 2.33c	3.43 ± 0.962ab	5.87 ± 2.47c	2.96 ± 1.62a
Ni	81 ± 43.5b	26.3 ± 15.1a	27.5 ± 12.6a	28.6 ± 9.70a	29.7 ± 10.9a	22.6 ± 8.08a	73 ± 38.7b	23.3 ± 10.3a
Cu	170 ± 78d	220 ± 126e	159 ± 69cd	112 ± 37.9ab	150 ± 95bcd	94 ± 59a	97 ± 65a	120 ± 63abc
Zn	724 ± 347e	288 ± 114a	378 ± 131ab	394 ± 168b	534 ± 243cd	424 ± 135b	597 ± 203d	448 ± 239bc
As	12.2 ± 9.20d	6.21 ± 2.19bc	5.94 ± 2.51abc	5.86 ± 2.09abc	4.02 ± 1.74a	4.43 ± 1.27ab	6.70 ± 6.01c	4.33 ± 2.72ab
Mo	2.13 ± 1.12b	3.43 ± 1.84c	1.91 ± 1.10ab	2.12 ± 1.15b	1.92 ± 1.69ab	1.52 ± 1.39ab	1.36 ± 0.578a	1.45 ± 0.990a
Cd	1.35 ± 0.434e	0.205 ± 0.099a	0.334 ± 0.209a	0.354 ± 0.287ab	0.516 ± 0.221bc	0.285 ± 0.147a	1.11 ± 0.535d	0.663 ± 0.535c
Pb	20.6 ± 7.39d	6.86 ± 1.78a	11.4 ± 4.08b	11.0 ± 4.94b	17.7 ± 6.30c	13.5 ± 3.76b	25.2 ± 9.82e	13.6 ± 7.71b
Al	1058 ± 552d	664 ± 403ab	820 ± 377bc	867 ± 658bcd	1055 ± 592d	758 ± 222ab	990 ± 293cd	553 ± 484a
Se	12.7 ± 0.602e	10.1 ± 1.75d	9.40 ± 1.24c	9.75 ± 1.76cd	8.48 ± 1.18a	8.55 ± 1.21ab	9.15 ± 1.22bc	8.36 ± 1.21a

Notes: NE: Northeast; XJ: Xinjiang; HM: Helan Mountain; HC: Hexi Corridor; LP: Loess Plateau; YV: Yanhuai Valley; BG: Bohai Gulf; SWH: Southwest Highland. All results larger than the linear range were calculated after diluting different multiples. The lowercase letters a–e indicate significant variation at the *p* < 0.05 level. SD = standard deviation.

**Table 3 molecules-24-00248-t003:** Estimated daily intake (EDI, μg/kg bw/day) of each element via consumption of wine from eight Chinese regions.

Region ^1^	Cr		Mn		Ni		Zn		As		Mo		Cd		Pb		Al		Se	
	Mean	P95	Mean	P95	Mean	P95	Mean	P95	Mean	P95	Mean	P95	Mean	P95	Mean	P95	Mean	P95	Mean	P95
NE	0.704	0.821	26.0	30.7	0.270	0.325	2.41	2.85	0.041	0.052	0.007	0.008	0.004	0.005	0.069	0.078	3.53	4.21	0.042	0.043
XJ	0.460	0.485	6.09	6.54	0.088	0.102	0.96	1.07	0.021	0.023	0.011	0.013	0.001	0.001	0.023	0.025	2.21	2.59	0.034	0.035
HM	0.437	0.462	6.23	6.65	0.092	0.102	1.26	1.37	0.020	0.022	0.006	0.007	0.001	0.001	0.038	0.041	2.74	3.06	0.031	0.032
HC	0.466	0.498	6.55	7.17	0.095	0.108	1.31	1.52	0.020	0.022	0.007	0.009	0.001	0.002	0.037	0.043	2.89	3.71	0.032	0.035
LP	0.443	0.483	7.43	8.10	0.099	0.116	1.78	2.16	0.013	0.016	0.006	0.009	0.002	0.002	0.059	0.069	3.52	4.44	0.028	0.030
YV	0.410	0.436	6.87	7.57	0.075	0.084	1.41	1.57	0.015	0.016	0.005	0.007	0.001	0.001	0.045	0.049	2.53	2.78	0.029	0.030
BG	0.503	0.528	20.8	24.1	0.242	0.283	1.99	2.21	0.022	0.029	0.005	0.005	0.004	0.004	0.084	0.094	3.30	3.61	0.030	0.032
SWH	0.393	0.424	7.61	8.32	0.078	0.088	1.49	1.72	0.014	0.017	0.005	0.006	0.002	0.003	0.045	0.053	1.84	2.30	0.028	0.029
PTDI			183 ^2^		12.0		1000		2.14 ^3^				0.833 ^4^				143 ^5^		6.67 ^6^	

^1^ NE: Northeast; XJ: Xinjiang; HM: Helan Mountain; HC: Hexi Corridor; LP: Loess Plateau; YV: Yanhuai Valley; BG: Bohai Gulf; SWH: Southwest Highland. ^2^ The PTDI (μg/kg bw/day) of Mn was calculated from a tolerable daily intake of 11.0 mg/day by a 60 kg adult. ^3^ The PTDI (μg/kg bw/day) of As was calculated from a provisional tolerable weekly intake (PTWI) of 15 μg/kg body weight. ^4^ The PTDI (μg/kg bw/day) of Cd was calculated from a provisional tolerable monthly intake (PTMI) of 25.0 μg/kg body weight based on a month of 30 days. ^5^ The PTDI (μg/kg bw/day) of Al was calculated from PTWI of 1000 μg/kg body weight. ^6^ The PTDI (μg/kg bw/day) of Se was calculated from an upper tolerable limit for selenium of 400 μg/day by a 60 kg adult.

**Table 4 molecules-24-00248-t004:** Target hazard quotient (THQ) of trace elements from wines of different regions.

Elements	NE		XJ		HM		HC		LP		YV		BG		SWH	
	Mean	P95	Mean	P95	Mean	P95	Mean	P95	Mean	P95	Mean	P95	Mean	P95	Mean	P95
Cr	0.235	0.274	0.153	0.162	0.146	0.154	0.155	0.166	0.148	0.161	0.137	0.145	0.168	0.176	0.131	0.141
Mn	0.186	0.220	0.044	0.047	0.044	0.048	0.047	0.051	0.053	0.058	0.049	0.054	0.148	0.172	0.054	0.059
Ni	0.014	0.016	0.004	0.005	0.005	0.005	0.005	0.005	0.005	0.006	0.004	0.004	0.012	0.014	0.004	0.004
Zn	0.008	0.009	0.003	0.004	0.004	0.005	0.004	0.005	0.006	0.007	0.005	0.005	0.007	0.007	0.005	0.006
As	0.137	0.173	0.070	0.077	0.067	0.073	0.067	0.073	0.043	0.053	0.050	0.053	0.073	0.097	0.047	0.057
Mo	0.001	0.002	0.002	0.003	0.001	0.001	0.001	0.002	0.001	0.002	0.001	0.001	0.001	0.001	0.001	0.001
Cd	0.004	0.005	0.001	0.001	0.001	0.001	0.001	0.002	0.002	0.002	0.001	0.001	0.004	0.004	0.002	0.003
Se	0.008	0.009	0.007	0.007	0.006	0.006	0.006	0.007	0.006	0.006	0.006	0.006	0.006	0.006	0.006	0.006
Total	0.593	0.708	0.284	0.306	0.274	0.293	0.286	0.311	0.264	0.295	0.253	0.269	0.419	0.477	0.250	0.277

Notes: NE: Northeast; XJ: Xinjiang; HM: Helan Mountain; HC: Hexi Corridor; LP: Loess Plateau; YV: Yanhuai Valley; BG: Bohai Gulf; SWH: Southwest Highland.

**Table 5 molecules-24-00248-t005:** The varieties and number of 315 wines in different regions.

Region	Varieties	Number of Wines
NE	Beibinghong, Vidal	30
XJ	Cabernet Sauvignon, Merlot	50
HM	Cabernet Sauvignon, Merlot	60
HC	Cabernet Sauvignon, Merlot	30
LP	Cabernet Sauvignon, Cabernet Franc	20
YV	Cabernet Sauvignon, Syrah	35
BG	Cabernet Gernischet, Marselan	40
SWH	Rose Honey, Crystal	50

Notes: NE: Northeast; XJ: Xinjiang; HM: Helan Mountain; HC: Hexi Corridor; LP: Loess Plateau; YV: Yanhuai Valley; BG: Bohai Gulf; SWH: Southwest Highland.

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
