# Peer review of "A Human Health Risk Assessment of Trace Elements Present in Chinese Wine"

_molecules, 2019, doi:10.3390/molecules24020248_

Round 1
Reviewer 1 Report
The presented manuscript by Fang et al. describes a study to determine the heavy metals content in Chinese wines by ICP-MS. ICP-MS for metal trace analysis in wine samples is a standard and well-established method. Similar reports have been published for other wine regions through the world. The originality of this study to the analytical chemistry readers is questionable. I also have the following concerns.
1. The wines from eight wine-producing regions were used to study the geographical feature of heavy metals in wines. But are the chosen wine samples representing each wine-producing regions? Are the wines samples selected from different vineyards and different winery in each regions? How many vineyards and winery?
2. One objective of this study is to “establish and validate an analytical method for the determination of trace elements in wines (Page 2, line 87-88)”. But the validation results were not presented or discussed in the manuscript except a very brief sentence in quality control section. Please present the validation data in the manuscript and provide detail discussion about the validation experiment and method performance based on the validation result, such as the spike levels, spike recovery, precision and accuracy of the certified reference material.
3. The linear range of calibration curve were 10-250 ug/L for Al and 0.25-50 ug/L for other heavy metal elements (Page 4, line 147-148). However, some result in table 2 were out of the calibration curve range. Among the out of range result, the Cd concentration in XJ wines were below LOQ, others were above the linear range. How were they determined?
4. Page 8, line 250-252. The authors attributed the high levels of heavy metal in DB wines to the wine type (ice wine) as a more concentrated wine. This is introducing another variable other than the wine-producing region. I suggest to have separate wine-producing region comparison for the regular wines and the more concentrated wines.
5. Please elaborate “… difference in Cu content might be due to the use of different grape cultivation methods such as the use of Bordeaux” (Page 8, line 261-262). How is the cultivation method affecting the Cu level?
Author Response
Point 1: The wines from eight wine-producing regions were used to study the geographical feature of heavy metals in wines. But are the chosen wine samples representing each wine-producing region? Are the wines samples selected from different vineyards and different winery in each region? How many vineyards and winery?
Response 1: Thanks for this question. We are very sorry for our negligence of the detailed explanation for these. We chose the wine samples that have the Protected Geographical Indication (PGI), and all samples are collected from 71 local different chateaus and wineries. Now they are added to the article in Page 3, line 114-116.
Point 2: One objective of this study is to “establish and validate an analytical method for the determination of trace elements in wines (Page 2, line 87-88)”. But the validation results were not presented or discussed in the manuscript except a very brief sentence in quality control section. Please present the validation data in the manuscript and provide detail discussion about the validation experiment and method performance based on the validation result, such as the spike levels, spike recovery, precision and accuracy of the certified reference material.
Response 2: Thanks for this suggestion. We are very sorry for our negligence of the detailed validation results. We have re-added and revised the content of verification methods and results, and now it is shown in Page 5, line 158-167, line 187-200 and Table 2.
Point 3: The linear range of calibration curve were 10-250 ug/L for Al and 0.25-50 ug/L for other heavy metal elements (Page 4, line 147-148). However, some result in table 2 were out of the calibration curve range. Among the out of range result, the Cd concentration in XJ wines were below LOQ, others were above the linear range. How were they determined?
Response 3: Thanks for this question. We are very sorry for our negligence of the detailed explanation for determination results. For the results beyond the linear range, our previous determination was to dilute wine samples in different multiples to the linear range, and then calculated the corresponding results. Now, it is shown in the notes of Table 3. Although some of the results are lower than LOQ in supplementary material, all the results are larger than LOD. The results we use are clearly presented by the instrument, and now the LOD and LOQ of each element are shown in Table 2, its definition is shown in Page 5, line 190-194.
Point 4: Page 8, line 250-252. The authors attributed the high levels of heavy metal in DB wines to the wine type (ice wine) as a more concentrated wine. This is introducing another variable other than the wine-producing region. I suggest to have separate wine-producing region comparison for the regular wines and the more concentrated wines.
Response 4: Thanks for this suggestion. It is really true as suggested that need separate wine-producing region comparison for the regular wines and the more concentrated wines. However, the main product of the Northeast wine producing region in China is ice wine, the wine samples we selected are representative of this region and it has the Protected Geographical Indication (PGI). In addition, what we have done is to assess and predict the risk of wine in this region, not be aimed at to the type and variety of wine.
Point 5: Please elaborate “… difference in Cu content might be due to the use of different grape cultivation methods such as the use of Bordeaux” (Page 8, line 261-262). How is the cultivation method affecting the Cu level?
Response 5: Thanks for this question. We are very sorry for our negligence of the details of the effect of Bordeaux on Cu. The main component of Bordeaux solution was basic cupric sulphate, and the demand of Bordeaux solution for vineyards varied from region to region, so the different amount of Bordeaux solution used in different regions led to the difference of copper concentration level. Now, we have added these in Page 11, line 297-300.
Special thanks to you for your good comments.
Reviewer 2 Report
please see my comments attached

Author Response
Point 1: The revised manuscript presents a study reporting an analytical survey of trace element concentrations in 315 Chinese commercial wines from eight winegrowing regions to assess their human health risk from their daily intake. The topic is interesting. However, I am not left fully convinced that there is enough material for a ‘research article’. Nevertheless, I think the subject matter can be of interest to researchers and industry as the authors provided a large data set for the current situation in China. For this reason, I think the authors need to re-submit this study under a new ‘communication’ format rather than a ‘research article’, unless the Editors think the opposite. English also requires some editing from a native language speaker. There are a number of unclear sentences or wrong words used (L31: in the other hand; L40-41; L47-48; L52-54; L149-151; etc..).
Response 1: Thanks for this question. We are very sorry for our less detailed writing that let you misunderstand our research. First, the idea of our study is that through the accurate and effective determination of trace elements that are potential health risks in wine by our established method, using the methods and information found assesses possible hazards for drinking Chinese wine. The methods and materials used are mostly from the Joint FAO/WHO Expert Committee on Food Additives (JECFA) and the United States Environmental Protection Agency (US EPA), so we think this is a research article, not a communication. Similar studies include International Journal of Food Contamination volume 1 issue 1 (2014), Journal of Agricultural Chemistry and Environment 2 (2013) 35-41, Environmental Earth Sciences volume 76 issue 14 (2017) and Environmental Science and Pollution Research 23 (2016) 7794-7806, etc. Then, we regret there were problems with the English. The paper has been carefully revised by the American Journal Experts (AJE) to improve the grammar and readability. Now, the problems are corrected and presented in L32, L39-42, L48-50, L54-56, L163-165.
Point 2: Title: Please, replace ‘ingested’ with ‘present’. Abstract: L21-23: this sentence is not a real conclusion. Please, change it.
Response 2: Thanks for this suggestion. We are very sorry for our negligence of the usage of words and summary of conclusions. Now, we have replaced ‘ingested’ with ‘present’ in title, and the change of Abstract is shown in L21-23.
Point 3: Introduction: L27: ‘…wine from grapes and wine from grape juice’. Please, rephrase as wine is a grape juice fermenting product only. L28: wine is not the most popular beverage in western countries, water is. If the authors mean ‘alcoholic drink’, beer is the most popular, no wine. L41: ‘…accumulation of flavours’ is incorrect. Flavour is a sensory dimension not chemical. It should be changed to: ‘...development of aroma precursors responsible for wine flavour attributes…’. L42: replace ‘its fruits’ with ‘the grapes’. L43: The alcohol content is generally high because winemakers delay harvesting to ensure a fully development of aroma precursors. Delete large differences between night and daily temperature. This is also incorrect. L55-80: this is the section where the authors should provide more detail on the daily intake/regulations of these trace elements. Now, there are just some associations between element and diseases, but no official concentrations have been mentioned. This is necessary to provide a context of daily intake risk.
Response 3: Thanks for this suggestion. We are very sorry for our inaccuracy of wording and unclear comprehension of some knowledges. First, the changes of Introduction are shown in L27-28, L40-42, L43-45. Then, we have added some official concentrations in L63-89, and the references are also marked.
Point 4: One of the most important factors that influence the presence of trace elements in wine is the soil. Thus, the authors must provide much more information in regards (as supplementary material). Please, add a table presenting each element together with its m/z, detection mode, limit of detection (LOD). It is confusing now.
Response 4: Thanks for this suggestion. We are very sorry for our negligence of some information. Although one of the most important factors that influence the presence of trace elements in wine is the soil, there are many important factors also could influence, such as the production process and cultivation technology. Nevertheless, the target of our study is the wine products, and we consider not the effect of soil on trace elements in wine, but the safety assessment of wine by trace elements in wine. Then we have referenced researches include Biological Trace Element Research 153 (2013) 119-126, International Soil and Water Conservation Research 1 (2013) 65-75, Advanced Materials Research 726-731 (2013) 239-244 and Acta Ecologica Sinica 32 (2012) 1803-1810. However, there is no complete data on trace elements in soils of the studied areas especially related vineyards, and soil samples were not collected when wine samples were collected. Therefore, we cannot provide information about trace elements in soil. Then, we have added the detection mode that is standard mode in L156, the nuclide information as m/z and limit of detection (LOD) both are added in Table 2.
Point 5: L98-101: Those acronyms need to be changed. They do not make any sense now. E.g. ‘Bohai Gulf’ should be BG not HB; ‘North East’ should be NE not DB; ‘Loess Plateau’ should be LP not HT; ‘South West Highland’ should be SWH not YCZ. L132-135: Rephrase. Unclear. L108-139: specify manufacture of each chemical/reagent/piece of equipment properly, indicating: manufacture trade mark, city, country. It is inconsistent and confusing now.
Response 5: Thanks for this suggestion. We are very sorry for our negligence of the detailed description and the confusion caused by these. Now, we have replaced ‘HB’, ‘DB’, ‘HT’ and ‘YCZ’ with ‘BG’, ‘NE’, ‘LP’ and ‘SWH’, respectively, it is shown in full text. And the L132-135 has been changed to ‘Finally, the digested solution was constant volume to 25 mL with 2% nitric acid solution and waiting for determination. Reagent blanks and matrix spike duplicates were also treated in accordance with the abovementioned procedure’ in L146-149. Then, we have re-edited and reorganized the writing of manufactures, and they are shown in L122-153.
Point 6: L153-173: The authors must justify their choice to use a 60 kg adult drinker consuming 200 mL of wine per day as model. Why a 60 kg adult and 200 mL of wine per day? Please, extend.
Response 6: Thanks for this question. We are very sorry for our negligence of the detailed explanation for indexes used. First, there were no related data about the wine daily consumption rate of drinkers in China, and the target of our assessment was adult drinkers, so we assumed the volume of 200 mL (approximately a glass of wine) as the wine daily consumption rate of drinkers. Other reasons include the similar to other study and Chinese drinking habits. Then, because the smaller the body weight, the larger the EDI calculated, we selected 60 kg as the adult drinker average weight for calculations so that the assessment results could be applied to more people. They are shown in L313-320.
Point 7: L156-160: the authors analysed each wine in triplicates (analytical repetitions). So, no actual statistical analysis could be performed as stats can be performed on biological replicates, not on repeated measurements over the same or artificially generated experimental units. Please, revise the text and tables accordingly. There are some cases when you can apply ANOVA on repeated measurements, but only under specific experimental conditions no yours. The authors instead should apply some chemometric approach on the data set for pattern recognition to visualize group clustering of wine samples according geographic origin. This will make easier the visualization of differences between wines from different geographic locations. See Food Chemistry 164 (2014) 485–492 or Food Chemistry 133 (2012) 1081–1089 for a good example of data presentation.
Response 7: Thanks for this question. It is really true as suggested that some chemometric approach on the data set for pattern recognition to visualize group clustering of wine samples according geographic origin to make easier the visualization of differences. However, first, the triplicates of each wine mean the sampling the same wine three times from one wine barrel, not three times determination, we think these are effective statistical replicates due to the complex wine production process. Then, the purpose of the ANOVA used is comparing the differences of each trace element among different producing regions, and our analysis is based on the differences of elements concentration level between in each region, using the ANOVA is more objective and clearer. On the other hand, it is not necessary for our study to use cluster analysis because we have not done the identification and division of wine origin in this study.
Point 8: L161-173: I am confused. Did the authors make up those equations to calculate the risk for human health? Or were those equations sourced from relevant existing literature? How can they say that at THQ is <1, there are no obvious adverse effects? Or vice versa (THQ>1)? This part needs an extensive revision supported my relevant medical literature before those claims can be made. Figure 1: Remove ‘Chinese Wine Region’ from figure 1 with the red circle. Unnecessary. Please, also spell the acronyms out. There is no need to abbreviate those geographical locations in such big figure. Table 1: Replace ‘Amount’ with ‘Number of wines’. Remove the last row (Total…), you could add ‘315’ in the table title.
Response 8: Thanks for this suggestion. We are very sorry for our negligence of the detailed explanation and figure information. All equations and the explanation of THQ are derived from official guideline and related studies, now they are marked in references 24-27. Then, Figure 1 and Table 1 have been revised according to these suggestions, now they are shown in Figure 1 and Table 1.
Point 9: R&D: L176-330 & Tables 2-4: Check all text and tables for significant digits, for result expression. Decide whether you need so many decimal digits each time for results. Revise tables accordingly. Usually is for values ranging from 0 to 1: ‘3 decimals’; from 1 to 10: ‘2 decimals’; from 10 to 50: ‘1 decimal’; greater than 50: no decimals. It is completely confusing and arbitrary now. Tables 2-4: add a legend to each table for the geographic origin acronyms. How can a reader remember each single region without a legend? Be consistent when mentioning those geographical areas in the text. Sometimes, the authors use the acronym and sometimes the full name. Please, just use the full name between ‘Hexy Corridor’ everywhere. Those acronyms are hard to remember.
Response 9: Thanks for this question. We are very sorry for our negligence of the significance digit and acronym. We have re-edited the significance digit of results as values ranging from 0 to 1: ‘3 decimals’; from 1 to 10: ‘2 decimals’; from 10 to 50: ‘1 decimal’; greater than 50: no decimals. Then, we have added a legend for the geographic origin acronyms in Table 1 and Tables 3-5. Finally, we have replaced acronyms with the full name of geographical areas in the full text.
Special thanks to you for your careful review and good comments.
Round 2
Reviewer 1 Report
I appreciate the authors to provide additional supporting evidence and further discussion. I have read the responses carefully and the modifications to the manuscript. Overall, the quality of the revised manuscript is significantly improved. However, I still have comments for point 2 and point 3.
Point 2: Certified reference material (GBW 10010) was used as an important quality control practice in each batch. But the result was not presented in the manuscript. From the reader’s standpoint, the result of CRM provide straightforward understanding about the method performance in terms of accuracy and precision. So I suggest to include the test result of the CRM in the manuscript, section 3.1 method verification results.
Point 3: LOQ = limit of quantification. So technically speaking, the result below LOQ was not reliably determined. Although there were instrument reading, they no longer serve the purpose of accurate quantification. They are at most an estimated result. The author should indicate the < LOQ result as estimated result in the supplementary table S1.
Author Response
Response to Reviewer 1 Comments (Round 2)
Point 2: Certified reference material (GBW 10010) was used as an important quality control practice in each batch. But the result was not presented in the manuscript. From the reader’s standpoint, the result of CRM provides straightforward understanding about the method performance in terms of accuracy and precision. So, I suggest to include the test result of the CRM in the manuscript, section 3.1 method verification results.
Response 2: Thanks for this suggestion. We are very sorry for our negligence of the information of certified reference material. Now, we have added the determination results of GBW 10010 (certified reference material) in Table 2, and the explanation is given in line 196-198.
Point 3: LOQ = limit of quantification. So technically speaking, the result below LOQ was not reliably determined. Although there was instrument reading, they no longer serve the purpose of accurate quantification. They are at most an estimated result. The author should indicate the < LOQ result as estimated result in the supplementary table S1.
Response 3: Thanks for this suggestion. We are very sorry for our negligence of the explanation of results below LOQ. We have revised the format of values below LOQ in Table S1 and Table 2 that we have added ‘*’ as a mark and it is explained under the Table.
Thanks again to the reviewer for careful review and hard work.
Reviewer 2 Report
Please, see attached comments

Author Response
Response to Reviewer 2 Comments (Round 2)
Point 1: L12-13: Please change to: ‘... in 315 wines. Samples were…’. L27-28: Please delete: ‘Wine is one of….and history’ (confusing, and wrong, again). Start with: ‘In recent…’. L41 & 44: My original comment was uncompleted, I am sorry for that. Please, change to: ‘…development of grape aromas.’ (delete precursors, as aromas exist also as free volatile compounds not just bond to sugars.). Acronyms: in the first review report I gave an example to the authors how properly attribute an acronym to a geographical name. However, they have just corrected what I have suggested and left out the rest. For example, HL: Helan Mountain should be changed to HM; HX: Hexi Corridor should be changed to HC; YH: Yanhuai Valley should be changed to YV. Supplementary material: It is not really a reviewer job to specify every single instance where the change has to be made. But it looks like all legends in the supplementary figures (1-12) and Table S1 still need to be update with the new acronyms. Table S1: Please, make sure the first row with the metals acronyms appears in each page, so that readers can actually see to what each column corresponds.
Response 1: Thanks for these suggestions. We have read the suggestions carefully and made correction according to the comments. The changes are as follow: (1) We have adjusted the first sentence of abstract as ‘The concentrations of trace elements in wines and health risk assessment via wine consumption were investigated in 315 wines. Samples were collected from eight major wine-producing regions in China.’, and it is shown in L12-14. (2) We have deleted the first and second sentences of introduction, now it is starting with: ‘In recent…’ and shown in L27-28. (3) We have deleted the ‘precursors’ in L41 and L44, and now it is shown in L39 and L42. (4) We have replaced ‘HL’, ‘HX’ and ‘YH’ to ‘HM’, ‘HC’ and ‘YV’ respectively, and it is applied to the full manuscript. (5) We have updated all acronyms to the supplementary material. (6) We have added a header to the first line of each page in Table S1.
Point 2: It is still not clear (at least to me) how many “wines” and “samples” (wine ≠ sample) the authors have analysed. This is not specified as the authors sometimes use the word: ‘wine samples’, sometimes ‘wines’ and sometimes ‘samples’ in the table headings and text. This adds confusion to confusion. ‘Wines’ mean the tank or the barrel (e.g. Premium Estate Merlot 2015); ‘samples’ mean all 3 replicates; ‘wine samples’ mean aliquots of wines (e.g. 30 mL).
In view of this comment (extremely confusing by the way), what did the authors present in Table 3 and Supplementary Table S1? The authors show a Table as supplementary material (Table S1) with 315 rows, each one with a different denomination and a single ‘value’. The authors should clearly say what each row and denomination is? Are these ‘315’ different wines from ‘315’ different tanks/barrels? Or, are these ‘105’ different wines from ‘105’ tanks/barrels in triplicates (105 x 3= 315)?
Response 2: Thanks for these questions. We are very sorry for our negligence of the clear conceptual expression. The 315 wines we collected from different tanks or barrels did not contain duplicates. The results of each wine are the mean value of three duplicates. It is really true as suggested that ‘wine ≠ sample’, and we have replaced ‘wine samples’ to ‘wines’ in full text except in Sampling and Quality Control. The number of wines in Table 3 and Table S1 represents wines but not samples, and we have added the notes of the denomination of each row in Supplementary Table S1.
Point 3: The way the means (± SD), presented in Table 3, have been calculated is completely unclear, as well as the analysis of 1-way variance has been applied. As I have previously suggested, ANOVA can’t be applied. Please, see: Am. J. Enol. Vitic., Vol. 50, No. 4, 1999 (Table 3) as an example.
In particular, it still remains extremely confusing and vague how many wines were analysed, as well as the statistical approach used. In my opinion, there are still unsurmountable problems with the stats, which the authors did not seem to handle very well. To me, the authors MUST seek assistance from a professional statistician to find the best way to present their data, as well as highlight differences between geographical regions. Additionally, the authors did not highlight the changes that were made in the new revised manuscript, making this review extremely difficult and time consuming.
Response 3: Thanks for these questions. We are very sorry for the misunderstanding and confusion caused by our articles. Firstly, the significant variations of trace elements in different regions were analysed by Duncan’s multiple range test using SPASS software. Secondly, the means presented in Table 3 were calculated by the results of different numbers of wines in each region, not contain the results of duplicates, the results of three duplicates were shown in Table S1. Then, we cannot find the ‘Am. J. Enol. Vitic., Vol. 50, No. 4, 1999 (Table 3)’ because we don’t know which page and which article, so we cannot use it as an example. Finally, we have used statistical analysis methods to find the variations we need and applied them to analysis. Although other methods of statistical analysis may be better, we think what we use is more suitable for our study.
Thanks again to the reviewer for careful review and hard work.